# Nest Survival and Transplantation Success of *Formica rufa* (Hymenoptera: Formicidae) Ants in Southern Turkey: A Predictive Approach

**Ayhan Serttaş [1], Özer Bakar [2]** **, Uğur Melih Alkan [1], Arif Yılmaz [1], Halil İbrahim Yolcu [3] and Kahraman İpekdal [4],***

[1] Western Mediterranean Forestry Research Institute, Antalya Regional Directorate of Forestry Campus, Muratpaşa, 07010 Antalya, Turkey; aserttas@hotmail.com (A.S.); ugurmelihalkan@gmail.com (U.M.A.); arifyilmaz@ogm.gov.tr (A.Y.)

[2] Department of Actuarial Sciences, Hacettepe University Faculty of Science, Beytepe, 06800 Ankara, Turkey; ozerbakar@gmail.com

[3] Manavgat Vocational School Organic Agriculture Program, Akdeniz University, Manavgat, 07600 Antalya, Turkey; hiyolcu@akdeniz.edu.tr

[4] Ahi Evran University Faculty of Agriculture, Bağbaşı, 40100 Kırşehir, Turkey

* Correspondence: kipekdal@gmail.com

**Abstract:** Research highlights: *Formica rufa* is used widely for biocontrol in Turkish forests. Although ecological characteristics of red wood ant habitats are well known, the statistical significance of these characteristics and their effects on nest transplantation success are largely unknown. Having such knowledge on a local scale, however, can help to predict the success of a scheduled transplantation effort, and can prevent loss of time and money. Background and objectives: In the present study, we used nest transplantation data from southern Turkey to determine habitat parameters that have a significant impact on nest survival, and to investigate possibility of predicting transplantation success from habitat parameter data. Materials and methods: Algorithms of data mining are widely used in agricultural and forestry applications for a wide range of tasks. In the present study, we used descriptive statistics to summarize the transplantation profile according to six habitat parameters (altitude, aspect, canopy closure, landform, nest substrate, and slope). We also used classification, a data mining approach, with two of its methods (decision tree and naïve Bayes) to determine the most important habitat parameters for nest survival and predict nest transplantation success in southern Turkey. Results: We found that altitude, aspect, and canopy closure were the most important factors affecting transplantation success. We also show that classification methods can be used in not only classifying, but also predicting the success rate of future transplantations. Thus, we show that the possibility of success for a given area can be predicted when certain parameters are known. Conclusions: This method can assist biological control practitioners in planning biocontrol programs and selecting favorable spots for red wood ant nest transplantation.

**Keywords:** biological control; red wood ant; mound transfer; data mining; Decision Trees; Naïve Bayes

## 1. Introduction

Wood ants (*Formica rufa* group) are the predominant ant species naturally occurring in temperate coniferous forests in Europe, particularly in Georgia, Russia, and Turkey [1]. They are also a keystone species, as they alter the soil composition and nutrient flow, and thus have an impact on tree growth by dispersing seeds, engaging in a mutualistic relationship with aphids, preying on invertebrates, and competing with other predators [2–5]. Owing to their predatory nature, they have been recognized

as beneficial predators since the 19th century [6], and have been introduced as biological control agents into forests and agroecosystems in and outside of their native range [7–9]. Several studies have shown that naturally occurring or transplanted ants can reduce the numbers of forest pest insects during outbreaks, such as pine processionary moths *Thaumetopoea pityocampa* and *T. wilkinsoni* (Lepidoptera: Notodontidae), pine sawfly *Neodiprion sertifer* (Diptera: Diprionidae), spruce budworms *Choristoneura spp.* (Lepidoptera: Tortricidae), and great spruce bark beetle *Dendroctonus micans* (Coleoptera: Curculionidae), by predation or harassment [6,10–14] (for possible disadvantages, see [15,16]).

As all *F. rufa* transplantation programs in Turkey were either reported in the intra-institutional media of the Turkish General Directorate of Forestry (OGM) or bulletins in the Turkish language, most of the related studies have remained as grey literature. Transplantation of *F. rufa* nests in Turkey was initiated in the first half of the 20th century by the OGM. According to the OGM reports, a total of 12,916 nests were transplanted interregionally between 1941 and 2018 in Turkey. The first scientific report on the issue belongs to Besçeli and Ekici [13], who addressed seven nests that were transplanted from central to southern Turkey. Next, Demirekin and Kolsuz [17] reported 500 nest transplantations conducted between 1970 and 1974, again from central to southern Turkey. They found that the number of transplanted nests decreased to 90 in 1998. However, the authors did not attempt to explain the possible reasons of this decrease. Öçal [18] found active nests mostly in pure cedar forests between 1377 and 1803 m altitudes in the Isparta region (northwestern Turkey). She showed that canopy closure level 2, a tree stump as a nest substrate, and 15–26° slopes were the habitat parameters that were preferred the most by the red wood ants as nest building grounds. She reported the nest frequency as 5.87 per hectare. In the Antalya region, located in further south, Serttaş et al. [18] determined the following parameters as the most favorable for nests: 1751–1800 m altitudes; east, north, and northeast aspects; canopy closure level 2; 21–30% slope; and fallen trunk as the nest substrate. Aksu and Çelik Göktürk [19] tried several nest transplantations in the Artvin region (northeastern Turkey), where they reported that success depended mainly on the compatibility of forest types and altitudes in the nest origin and in the transplantation site. They found that transplantation from Scots pine to spruce stands were not successful, and that an abundance of tree stumps in the transplantation sites had a positive impact on transplantation success. Finally, Avcı et al. [20] reported altitude, aspect, canopy closure, and tree species as the important parameters for transplantation success.

Apart from the studies from Turkey introduced above, several other studies, the majority being from northern Europe, have also focused on the parameters that can be responsible for nest survival. Among these parameters, altitude and slope [21], aspect [22], competition [23], disturbance [24], light [25,26], nest volume [11], humidity, and vegetation and soil characteristics [27–29] seem to have major impacts. Thanks to all of these studies, we have a sound understanding of red wood ant ecology, but we still do not have robust local scenarios that can lead help biological management decision-makers and practitioners towards to make more efficient nest transplantation more efficient. In order to produce such scenarios, the possible cumulative impacts of the habitat parameters that can be responsible in nest survival should be evaluated in local scales.

Data mining is the method for extracting information for the use of learning patterns and models from large datasets. It involves the uses of machine learning, statistics, artificial intelligence, database sets, pattern recognition, and visualization [30]. Classification is the most recognized modelling mechanism in data mining. It is a type of prediction problem and a process of assigning inputs to the target classes by a classifier (model) [31]. Decision tree, one of the most frequently used classification methods, is successful in predicting and explaining the relationship between features and their target value. It has several advantages, such as being nonparametric, not requiring any prior distribution assumption about input data, being able to work with any data type (nominal, numeric, and even text), and ease of use [32,33]. Classification can also be realized through a Bayesian approach. Naïve Bayes is a probabilistic algorithm using Bayes' theorem. Although it assumes independent parameters, it effectively solves a wide range of machine learning problems [34].

The present study is the first to test red wood ant nest survival and transplantation success statistically, and to evaluate different transplantation scenarios through decision tree and naïve Bayes methods at a local scale, in order to determine the most suitable places for transplantation. Accordingly, we asked the following questions: (1) what is the distribution of the considered nests, according to habitat parameters in the study areas? (2) Is there any difference in habitat parameters between active (*F. rufa* activity can be observed) and abandoned (no *F. rufa* activity), or between successful (active or passive but budding an active nest) and unsuccessful (passive and no budding, see below) transplantations? (3) Which parameter combinations are the best for nest transplantation in the considered region?

## 2. Materials and Methods

### 2.1. Sampling and Measurements

This study focused on a total of 615 red wood ant nests (468 transplanted, 147 newly formed) in Antalya and Isparta forests in south western Turkey (Figure 1). Data related to the nests in Isparta (160 transplanted, 82 new) were collected between 2016 and 2018. We used data from Serttaş [18] for the nests in Antalya (308 transplanted, 65 new). Red wood ants were absent in both regions prior to transplantation. Therefore, we could determine new nests budded from the old ones by comparing coordinates taken during original transplantation efforts, as well as those taken during current field observations for the study presented here and by Serttaş [18].

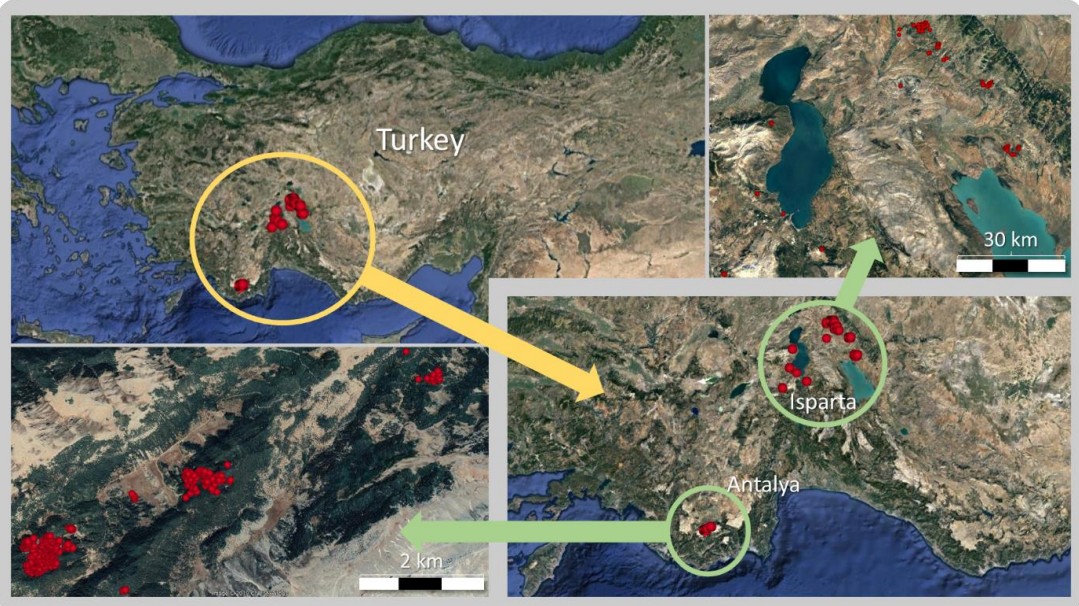

**Figure 1.** Study sites in Antalya and Isparta, Turkey.

After determining the nest survival (active/abandoned), we measured the following habitat parameters: altitude, aspect, canopy closure (0: crown closure less than 10%; 1: between 11% and 40%; 2: between 41% and 70%; 3: between 71% and 100%), landform (lower hill: the lowest 30% part of a hillside; mid-hill: 40% part in the middle of a hillside; upper hill: the highest 30% part of a hillside; ridge: apex of a hill; plain: flat land that is not part of a hill—these classes were determined separately for each topographic case through observation by the same person), nest height, nest radius, nest substrate (fallen trunk, tree stump, or open), and slope (sensu stricto [18]). Destroyed nests or nests with no ant activity were determined to be abandoned nests. Although protective cages or fences were placed as a part of the transplantation routine (Figure 2), some of the abandoned nests were destroyed. As the habitat parameters are expected to act collectively on nest survival, we had to consider the same

parameter sets for each nest. Therefore, we discarded nest height and nest radius parameters from the analyses. We thus used six habitat parameters: altitude, aspect, canopy closure, landform, substrate, and slope. However, we provided the entire dataset, including height and radius, toat ResearchGate. We also measured the pH of the soil samples taken at 0–30 cm and 30–60 cm depth from each nest point. Soil analyses were conducted at the Soil Laboratory of the Western Mediterranean Forestry Research Institute in Antalya, Turkey.

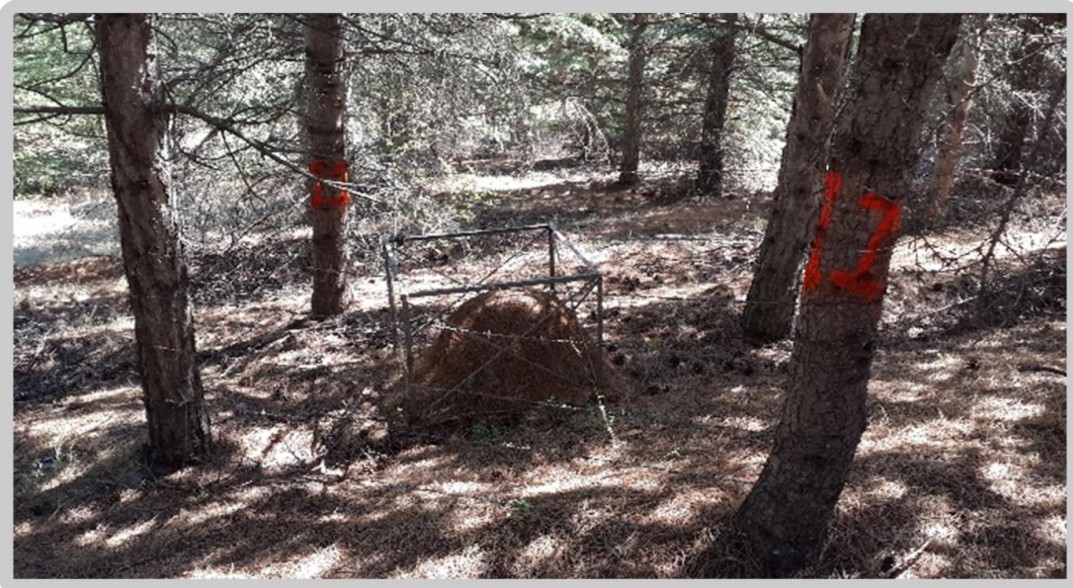

**Figure 2.** Transplanted *Formica rufa* nest with a protective cage in Isparta, Turkey (transplanted in 2012, photographed in 2016).

## 2.2. Descriptive Statistics

We determined the number of active and abandoned transplanted nests, along with new nests budded naturally after transplantation. We also counted the number of nests in each habitat parameter class (e.g., numbers of nests in stands with canopy closure level 0, 1, 2, and 3). We used these numbers to calculate active, abandoned, and new nest percentages in each parameter category. In order to compare active and abandoned nests for the measured parameters and pH measurements of different nest points, we applied Pearson's chi-square test.

We also determined the number of successful and unsuccessful nests and their counts in each habitat parameter class. Although an active nest could always be accepted as a successful transplantation, this was not true for all abandoned nests, due to the fact that a colony can abandon a nest to make a new one. In this case, transplantation should still be accepted as successful. Therefore, we reorganized the data set by using the following success criteria:

i.　　If a transplanted nest is active, transplantation is successful;

ii.　　If a transplanted nest is abandoned, but budded a new active nest, transplantation is successful;

iii.　　If a transplanted nest is abandoned and no new nest is observed, transplantation is unsuccessful;

iv.　　If a transplanted nest and the new nest budded from it are both abandoned, transplantation is unsuccessful.

In some cases, we determined that two transplanted nests that were close to each other became abandoned, and one new nest was founded in the same area. As red wood ants can merge separate nests and make a single nest, we accepted both transplanted nests as successful.

In order to compare successful and unsuccessful transplantations for the measured parameters, we applied Pearson's chi-square. We performed all statistical analyses in R [35].

### 2.3. Data Mining: Decision Tree and Naïve Bayes

We used the data set reorganized according to the success criteria described above. For the subsequent analyses, we applied a 70:30 train/test split to the data. In the present study, we used two different classification methods. The first method was the decision tree (DT) method, which works with an "if else–like" structure. It detects the most explanatory part in the complex data, and produces predictions based on this part. Therefore, predictions from DT analysis include information from data categories determined by the analysis. DT works with several algorithms, among which we selected the most frequently used ones—namely, C&RT (Classification and Regression Tree), CHAID (chi-squared automatic interaction detector), QUEST (quick, unbiased, efficient, statistical tree), and C5.0 (see [36] for further details). DT algorithms perform classification according to different criteria, and thus each algorithm can end up with different results. The algorithm that best suits the target variable and gives the best performance should be used. Thus, their performances must be tested. After determining which algorithms are to be used in the analyses, we used performance criteria to test the performance of each algorithm and to find the one that best suits the target variable. The performance criteria that we used were confusion-matrix-based accuracy, precision, sensitivity, specificity, F1 score, and area under the curve (AUC). A confusion matrix is the most frequently used method for measuring the performance of data with a predetermined target variable. This matrix, consisting of positive and negative predictions, shows how values in test data cluster with model values. "Accuracy" is the ratio of the number of correct predictions to the total number of predictions. "Precision" is the ratio of true positive predictions to total positive predictions. "Sensitivity" is the ratio of true positives to actual positives. "Specificity" is the ratio of true negatives to actual negatives. "F1 Ssore" is the harmonic mean of precision and sensitivity. Finally, AUC is the area under the curve in the function established between true and false positives. A detailed introduction of these performance criteria can be found in Sokolova and Lapalme [37]. Finally, we measured importance of habitat parameters in predicting the transplantation success by using DT built on the algorithms given above. These analyses were conducted by using IBM SPSS Modeler 18.2.

The second classification method that we used was naïve Bayes (NB). NB learns probabilistic relationships between the predictor and response parameters [38]. Unlike DT, NB scenarios include not only the explanatory part, but the entire data; therefore, resultant predictions from NB analysis include information from all data categories. We used NB to predict the probabilities of transplantation success based on the habitat parameters. This assumes that the parameters are independent. For this reason, dependent parameters, such as height and nest volume, should not be included together in the data in an NB analysis. As we already discarded these data for different reasons described above, we accepted the data set as independent. We also tested the performance of the algorithm through accuracy, precision, sensitivity, F1 score, and AUC criteria. We used Orange Data Mining software for the Bayesian analyses [39].

The two methods described above are different approaches to a data set, and both provide information related to it, all of which can be used for the prediction of, in our case, nest transplantation success. Therefore, these methods are not like experimental repeats or alternatives, but rather, they are more complementary to one each another.

## 3. Results

### 3.1. Transplantation Profile

We did not find any significant difference between the pH measurements of the soil samples from different localities and from different depths (pH = 6.3–7.7; between localities $p = 0.13$; between depths $p = 0.17$).

The majority of transplantations were at the highest interval (1701–1900 m), and transplantation frequency decreased in parallel with the altitude. The distribution of transplanted nests according to altitudes were as follows: 49.78% between 1701 m and 1900 m, 28.21% were between 1501 m and 1700 m,

16.88% between 1301 m and 1500 m, 4.49% were between 1101 m and 1300 m, and 0.64% were between 900 m and 1100 m (Figure 3a–i). The aspects where the majority of the transplantations were done were east, northeast, and north (73%, 73%, and 61%, respectively), while the least transplanted aspects were west and southwest (28% and 26%, respectively) (Figure 3a-ii). In total, 63% of all transplanted nests were in stands with the canopy closure level 2 (moderate closure). The percentage of transplanted nests at each of the other canopy closure levels was 12% (Figure 3a-iii). We found that mid-hill was the most preferred landform for transplantation (40.81%), followed by lower hill, ridge, upper hill, and plain (21.58%, 14.10%, 12.18%, and 11.32%, respectively) (Figure 3a-iv). The most preferred slope interval for transplantation was 21–30° (27.41%), followed by 31–40°, 0–10°, 11–20°, 41–50°, 51–60°, and 61–70° (19.70%, 18.63%, 16.70%, 10.28%, 5.35%, and 1.93%, respectively) (Figure 3a-v). Finally, we found that 53.85% of all transplanted nests were on tree stumps, 39.74% were on fallen trunks, and 6.41% were on open substrates (Figure 3a-vi).

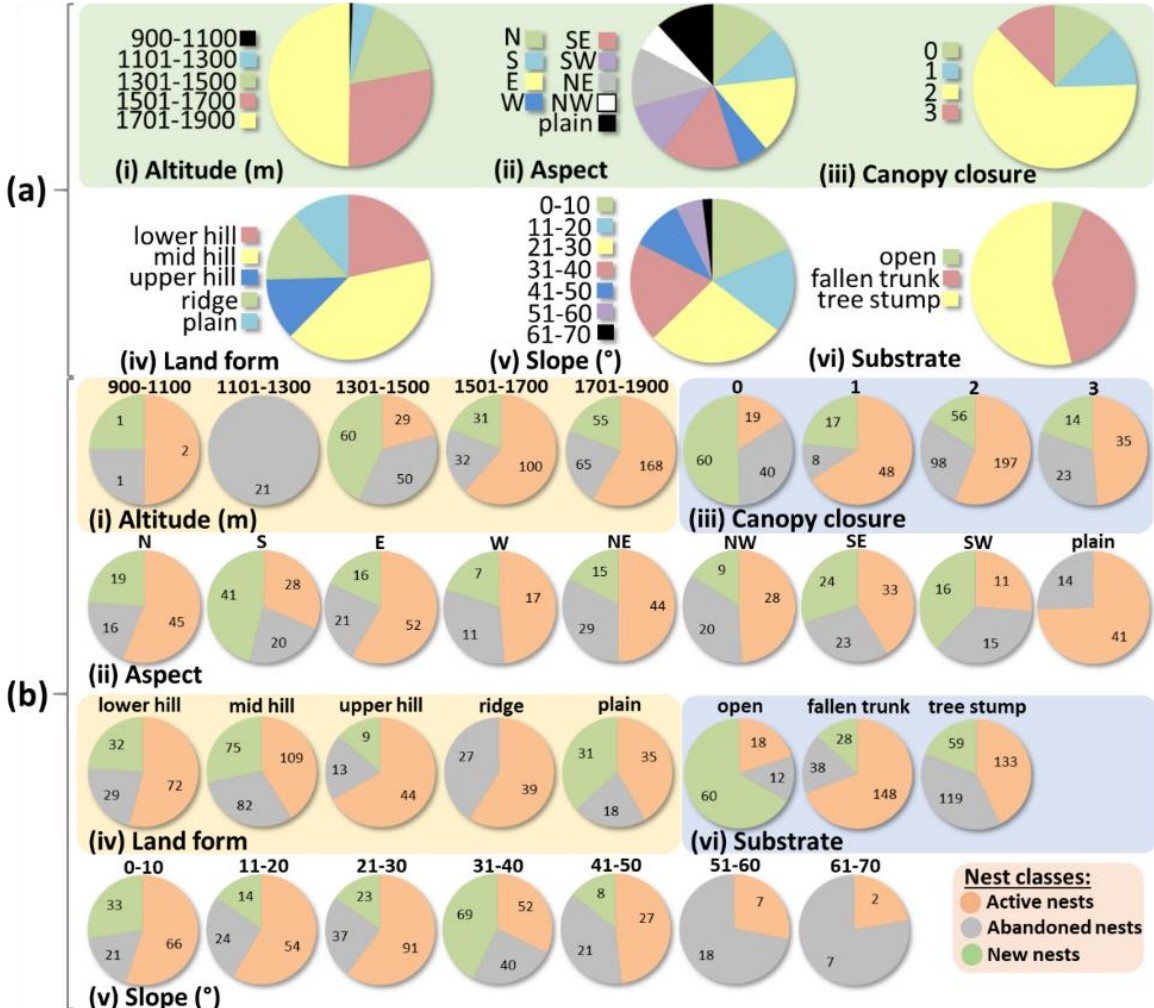

**Figure 3.** Distribution of *Formica rufa* nests according to different parameters studied. (**a**) Relative abundance of nests in different altitude (**a-i**), aspect (**a-ii**), canopy closure level (**a-iii**), landform (**a-iv**), slope (**a-v**), and substrate (**a-vi**) classes. (**b**) Active, abandoned, and new nest percentages in different altitude (**b-i**), aspect (**b-ii**), canopy closure level (**b-iii**), landform (**b-iv**), slope (**b-v**), and substrate (**b-vi**) classes. The numbers in the pie charts correspond to the number of nests in each nest class.

Among the transplanted nests, the numbers of active and abandoned nests were 299 and 169 (63.9% and 36.1%), respectively. We observed that 54 active nests produced new nests by budding (18%). We also found that 40 abandoned nests formed new nests (23.7%). Among the new nests,

the numbers of active and abandoned nests were 144 and 3 (98% and 2%), respectively. When both transplanted and new nests were considered together, we found that the ratio of abandoned nests was 28%. We compared the habitat parameters for active and abandoned nests, and found that they differ significantly in all parameters measured ($p < 0.00$), except for aspect ($p = 0.28$) and landform ($p = 0.02$). The highest ratio of active nests were between 1501 and 1700 m altitudes (75.76% of all nests were found to be transplanted to this interval), in plain aspect (74.55%), in canopy closure level 1 (85.71%), on upper hills (77.19%), between 0° and 10° slopes (75.86%), and on fallen trunks (79.57%). In contrast, the highest ratio of abandoned nests were between 1101 and 1300 m altitudes (100%), in the southwest (57.69%), in canopy closure level 0 (67.80%), on mid hills (42.93%), between 61° and 70° slopes (77.78%), and on tree stumps (47.22%). As the number of new abandoned nests were very low (3) compared to new active nests (144), we did not statistically compare the difference in their habitat parameters. On the other hand, we calculated new nest formation ratios for each parameter, and found the highest ratios between 1301 and 1500 m altitudes (75.95%), in the south aspect (85.42%), in canopy closure level 0 (101.69%), on plains (58.49%), between 31° and 40° slopes (75.00%), and on open substrates (200.00%) (Figure 3b).

*3.2. Transplantation Success Deduced from Descriptive Statistics*

By applying the success criteria (see Materials and Methods), we detected 338 successful and 130 unsuccessful transplantations. We then compared habitat parameters between successful and unsuccessful transplantations and found significant differences in all parameters ($p < 0.00$). Regarding altitude, most of the successful transplantations (58%) were between 1701 and 1900 m, whereas most of the unsuccessful transplantations (32%) were between 1301 and 1500 m. Among both successful and unsuccessful transplantations, northeast (aspect), level 2 (canopy closure), mid-hill (landform), and 21–30° (slope) were the parameter classes with the highest nest percentages (aspect: 32–15%; canopy closure: 66–55%; landform: 39–46%; and slope: 29–23% in successful and unsuccessful transplantations, respectively). In terms of substrate classes, we found the highest percentage of successful nests on fallen trunks (47%), whereas most of the unsuccessful transplantations were on tree stumps (76%) (Figure 4).

The highest number of transplantations was in the 1701–1900 m interval (Figure 3a-i). Figure 4 shows that the highest percentage of successful transplantations was also detected in this interval. The highest percentage of unsuccessful transplantations was recorded in the 1301–1500 m interval. Regarding aspect, canopy closure, landform, and slope, the classes having the highest number of transplantations were same as the classes having the highest percentage of successful and unsuccessful transplantations (northeast, level 2, mid-hill, and 21–30°, respectively) (Figure 3a-ii–v and Figure 4). The substrate class having the highest number of transplantations was the tree stump. The highest percentage of successful transplantations was in the fallen trunk class, whereas the highest percentage of unsuccessful transplantations was in the tree stump class.

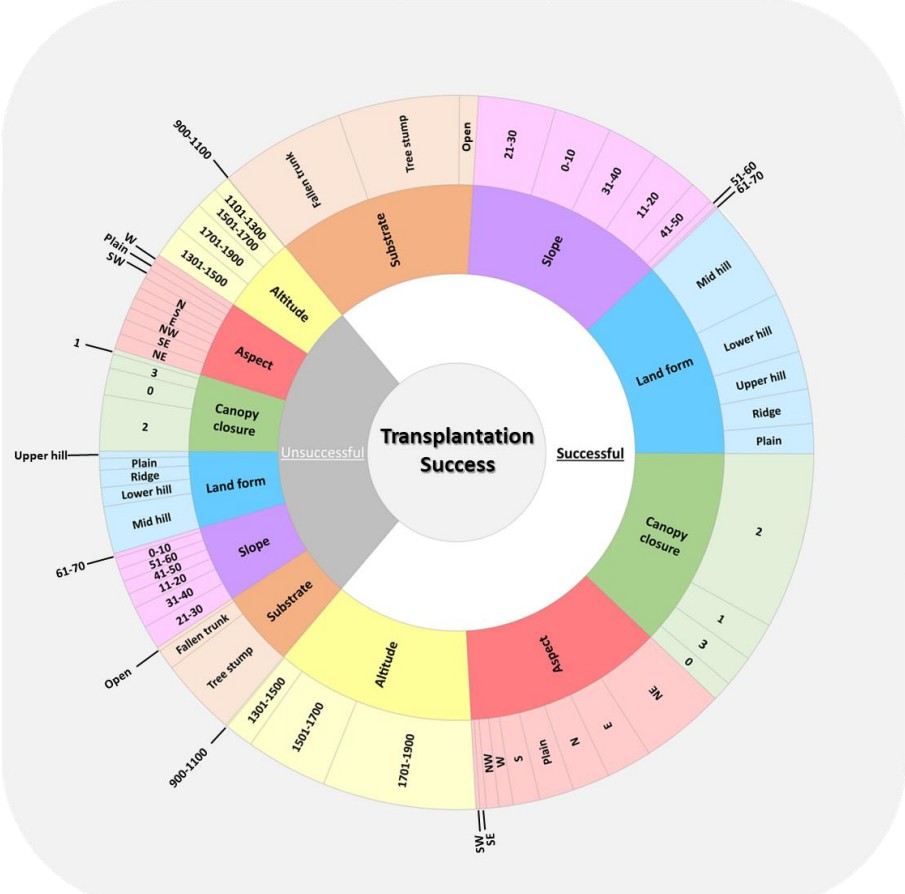

**Figure 4.** Distribution of *Formica rufa* nests according to success criteria (inner ring) and habitat parameters (middle and outer rings). The size of each sunburst slice corresponds to the number of nests it includes.

### 3.3. Transplantation Success Deduced from Data Mining

In general, all DT algorithms showed strong high performance in predicting transplantation success from the data. Therefore, all of them can be used for success prediction. In particular, the CHAID algorithm had the strongest highest performance under all performance criteria, except for the AUC criteria, under which the C5.0 algorithm had the strongest highest performance (Table 1). The most important habitat parameter in predicting transplantation success by using the DT algorithms was altitude. Aspect was the second most important parameter (Table 2).

**Table 1.** Performance measures for decision tree (DT) and naïve Bayes (NB) methods (highest scores are underlined).

| | DT | | | | NB |
|---|---|---|---|---|---|
| | **C&RT** | **CHAID** | **QUEST** | **C5.0** | |
| **Accuracy** | 0.734 | 0.822 | 0.767 | 0.815 | 0.763 |
| **Precision** | 0.745 | 0.852 | 0.791 | 0.844 | 0.759 |
| **Sensitivity** | 0.821 | 0.835 | 0.788 | 0.829 | 0.763 |
| **Specificity** | 0.546 | 0.591 | 0.497 | 0.579 | - |
| **F1 Score** | 0.780 | 0.813 | 0.753 | 0.806 | 0.761 |
| **AUC** | 0.591 | 0.668 | 0.671 | 0.671 | 0.757 |

**Table 2.** Importance percentages of habitat parameters in predicting transplantation success by the decision tree algorithms (highest scores are underlined).

|  | **C&RT** | **CHAID** | **QUEST** | **C5.0** |
|---|---|---|---|---|
| **Altitude** | <u>0.77</u> | <u>0.75</u> | <u>0.66</u> | <u>0.92</u> |
| **Aspect** | - | 0.17 | 0.09 | 0.08 |
| **Canopy closure** | 0.14 | - | 0.08 | - |
| **Landform** | 0.05 | 0.08 | 0.08 | - |
| **Slope** | - | - | - | - |
| **Substrate** | 0.04 | - | 0.09 | - |

Table 3 exemplifies the probabilities of transplantation success according to different conditions gathered through different algorithms. For example, the C&RT algorithm predicted the probability of success as 79.47% when the altitude was less than 1420 m and the slope was less than 57.5°. On the other hand, according to the QUEST algorithm, altitude alone was enough to get a certain rate of success. When the altitude was between 1398 and 1493 m, the predicted probability of success was 55% under the QUEST algorithm. On the other hand, changes in success predictions cannot be attributed to the altitude alone, as effects of other parameters given under the conditions in Table 3 are also important. It should be kept in mind that each result has its unique scenario, and they are not mutually exclusive; all can be used as different pieces of information related to the data.

**Table 3.** Examples of transplantation success predictions by decision tree and naïve Bayes methods. For each DT algorithm, there are NB conditions corresponding and adding to it. The cells tagged as (**ii**) and (**iii**) in the NB part show how probability changes after a minor change in the (**i**) cell.

| | | DT | | | NB | |
|---|---|---|---|---|---|---|
| | **Algorithm** | **Conditions** | **Probability of Success** | | **+ Conditions** | **Probability of Success** |
| **(a)** | C&RT | 1420 m < altitude slope < 57.5° | 79.47% | **(i)** | Aspect: north (N) Canopy closure: 3 Landform: upper hill Substrate: fallen trunk | 66% |
| | | | | **(ii)** | Aspect: east (E) Canopy closure: 2 | 78% |
| | | | | **(iii)** | Canopy closure: 1 | 87% |
| **(b)** | CHAID | 1371 m < altitude < 1609 m Aspect: E, N, west (W), northeast (NE), southeast (SE), southwest (SW) Canopy closure: 0 or 1 | 62.50% | **(i)** | Aspect: N Canopy closure: 0 Landform: upper hill Slope: 31–40° Substrate: fallen trunk | 70% |
| | | | | **(ii)** | Aspect: E | 74% |
| | | | | **(iii)** | Canopy closure: 1 | 95% |
| **(c)** | QUEST | 1398 m < altitude < 1493 m | 55.00% | **(i)** | Aspect: E Canopy closure: 0 Landform: upper hill Substrate: fallen trunk Slope: 11–20° | 76% |
| | | | | **(ii)** | Canopy closure: 3 | 90% |
| | | | | **(iii)** | Canopy closure: 1 | 95% |
| **(d)** | C5.0 | 1367 m < altitude < 1538.5 m Aspect: NE, NW, plain, south (S) | 74.28% | **(i)** | Aspect: NE Canopy closure: 1 Landform: upper hill Substrate: fallen trunk | 94% |
| | | | | **(ii)** | Aspect: NW | 94% |
| | | | | **(iii)** | Aspect: S | 94% |

In order to predict the success probabilities of different habitat parameter combinations, we conducted further NB analysis. The performance of this analysis was also high (Table 1). Unlike DT, NB used all parameters and produced success predictions for each parameter combination. As the number of possible parameter combinations were extremely high, it was not feasible to report them all here, and it was out of the scope of our focus. Therefore, we provided success prediction examples from DT and NB methods together, using several parameter combinations in Table 3, although it should be kept in mind that this is not an attempt to compare the two methods. Because they use different approaches, DT responded by selecting the best parameters, whereas NB responded using all parameters. According to the C&RT algorithm of DT, altitude lower than 1420 m and slopes lower than 57.5° together induced 79.47% success. In NB, keeping the same parameters and adding new ones, such as a northern aspect, canopy closure level 3, upper hill as the landform, and fallen trunk as the substrate, yielded in 66% success (Table 3a). Changing the aspect to east, canopy closure to 2, and keeping the other parameters the same made the probability of success 78% (Table 3a-ii). When canopy closure was changed to 1 in this parameter combination, the probability of success became 87% (Table 3a-ii). Examples for other algorithms, parameter combinations, and their success predictions can be seen at Table 3b–d.

We also reported three examples of high Bayesian success predictions in a Nomogram in Figure 5. Bayesian analyses also showed that the most important nest parameter for transplantation success was altitude. This was followed by canopy closure, substrate, slope, aspect, and landform. It should be noted here that NB classified continuous data (altitude and slope) according to their impact on the success. For example, 1526 m and lower altitudes were in the same class, as their contribution to success was not significantly different. Similarly, 39° and steeper slopes were in the same class. Increasing altitude had a positive impact on success, whereas the impact of increasing slope was negative. The nomogram in Figure 5a shows that probability of transplantation success was 91% when nest parameters were as follows: altitude = 1526–1699.5 m, canopy closure = 3, substrate = open" slope = 27–39°, aspect = north, and landform = upper hill. Similarly, when the altitude was higher than 1767.5 m, canopy closure level was 2, substrate was a tree stump, slope was less than 17.5°, aspect was east, and landform was upper hill, the probability of a successful transplantation was 80% (Figure 5b). When keeping the altitude and slope the same as in the previous example, but changing the canopy closure to 0, the substrate to fallen trunk, aspect to northeast, and landform to plain, the probability of success dropped to 70% (Figure 5c).

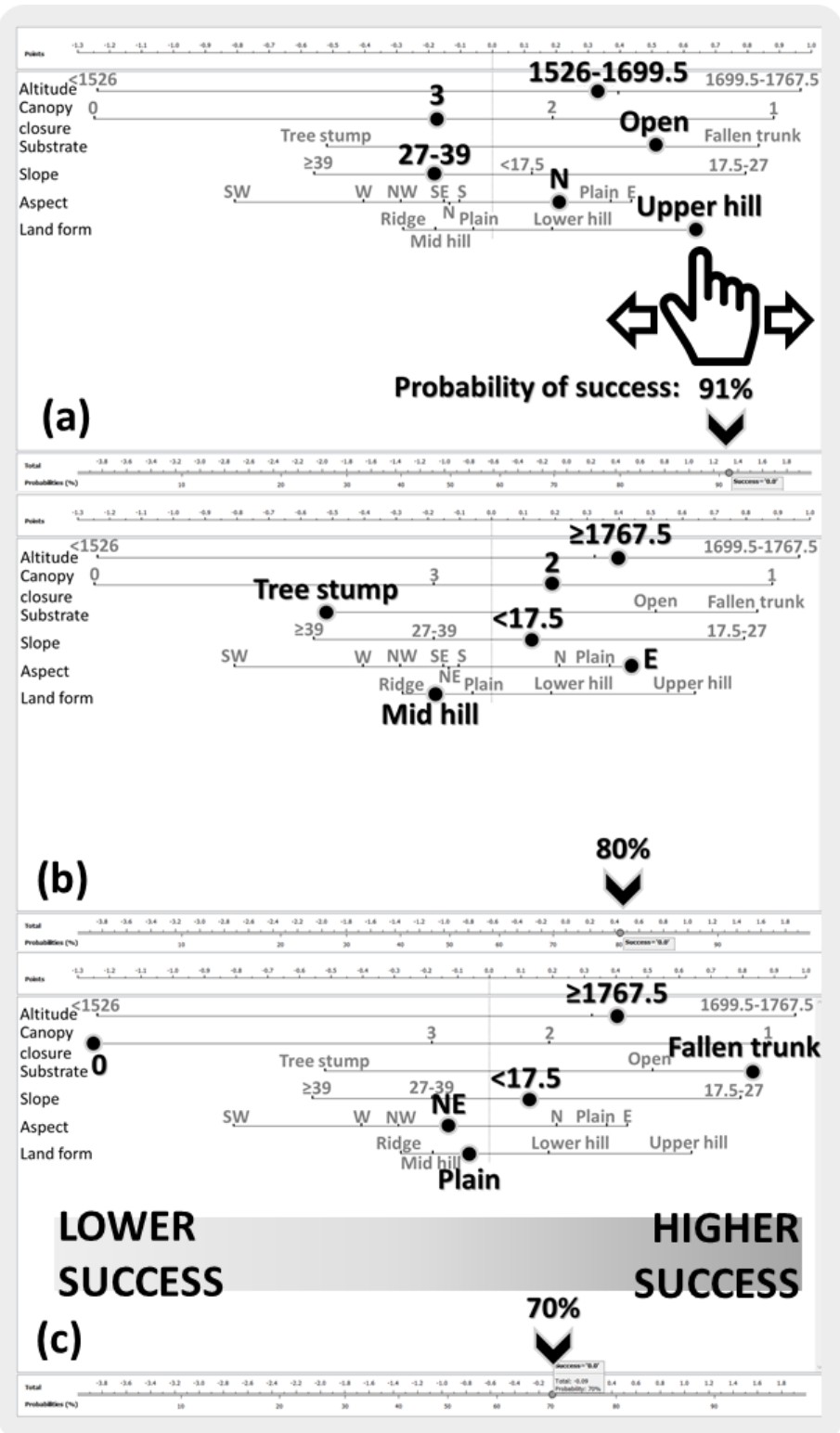

**Figure 5.** Three examples for nomogram output of naïve Bayes analysis for nest transplantation success prediction, predicting the success rate as 91% (**a**), 80% (**b**), and 70% (**c**). Parameters (altitude, canopy closure, substrate, slope, aspect, and landform) were sorted by the analysis, according to their importance in success prediction ("altitude" is the most and "land form" the least important parameter). Classes of each parameter were sorted by the analysis, according to their impact on success (from left: lower impact to right: higher impact).

## 4. Discussion

Red wood ants are widely used in Turkey to suppress forest pests. Although several studies have evaluated the factors that could be responsible in the successful transplantation of *F. rufa* nests, they lack robust statistical approaches that would help in decision-making at a local scale. In the present study, we tested and introduced an approach to evaluate the transplantation success of previous nest transplantation efforts, and to predict success in future applications. Data mining algorithms are widely used in agricultural and forestry applications for a wide range of tasks, from soil science to forest fire prediction (e.g., [40,41]) and from plant disease identification to the optimization of pesticide use (e.g., [38,42]). To the best of our knowledge, the study presented here is the first attempt to assist biological control practitioners in decision-making through the methods of data mining approach.

Klimetzek [43] found mean annual nest mortality for the three most abundant red wood ant species (*F. polyctena*, *F. pratensis*, and *F. rufa*) to be between 21% and 31%. Punttila and Kilpeläinen [44] reported that the percentage of abandoned red wood ant nests was 25% in Finland. We found a similar ratio (28%) when considered both transplanted and new nests together. Such close ratios over different geographic regions suggest that biological components (e.g., longevity of a colony) other than environmental parameters in nest survival may also be in action. We found that 23.7% of abandoned nests had formed new nests. This means that most of the former residents of the abandoned nests could not survive. On the other hand, many nests could disperse over long distances or merge into one nest, which might eventually cause an overestimation of the number of nests that failed to survive. Therefore, the rate of survivor colonies in our study could actually be higher. Finding the exact rate is only possible if a study dedicated to this question is planned during the transplantation process. We found that 18% of active nests budded to found new nests. High dispersal risks and low independent colony-founding success of individual wood ant queens are expected to promote a polygynous situation, wherein daughter queens are adopted into the queen nests. This strategy is often accompanied by the budding of the colony, whereby mated females of polygynous populations leave their natal nests with workers, and found a new nest in the vicinity of the ancestral nest [45]. In order to understand new nest formation and budding patterns, mating system (monogyny vs. polygyny) in the study region in particular should be investigated.

Altitude is one of the most important factors affecting *F. rufa* survival [46]. This seems to be even more evident in southern latitudes, as shown in the present study, and our findings related to altitude confirm the reports of Öçal [17], Serttaş et al. [18], and Avcı et al. [20]. It is not surprising to find most of the transplanted nests at high altitudes, as former Turkish foresters who conducted the transplantation efforts had already preferred high-altitude sites for transplantation. On the other hand, we found relatively more active nests at higher sites than lower sites. Accordingly, most of the successful transplantations were also at higher sites. This result was confirmed through descriptive statistics and the classification analyses (DT and NB). Considering that the native ranges of the nests transplanted to the study region are localities from northern Turkey, where the most significant difference from southern localities is lower mean temperature, higher transplantation success at higher southern altitudes is a result consistent with expectations.

Aspect is also among the factors influencing nest abundance in an area. Southern and western aspects are the most preferred aspects by red wood ants in most of the European sites [46]. On the contrary, we found more nests at eastern, northeastern, and northern aspects. This result confirms the report of Serttaş et al. [18]. Analyses of success showed that east, north, and northeast were the most favorable aspects for *F. rufa* in southern Turkey. This could be an adaptation to relatively higher summer temperatures in this region compared to the other European localities. The daily routine of red wood ants is related largely to ambient temperature. They are known to escape deep into the nests during the warmest hours of the day in summer [47]. This relationship between ant behavior and temperature, and the fact that *F. rufa* in the region was transplanted from cooler northern Turkey, suggest the possibility that northern aspects would be more favorable in warmer habitats.

Canopy closure is another significant habitat parameter related to red wood ant ecology [44]. Our results also confirm its importance. We found that most of the transplantations were made into stands with canopy closure level 2, which is the same level found in Öçal [17] and Serttaş et al. [18]. The highest percentage of active nests were in level 1, and the highest percentage of successful transplantations were in level 2. On the other hand, level 1 had the greatest and level 2 had the second greatest impact on transplantation success, according to the classification analyses. The highest ratios of abandoned nests were recorded in the canopy closure levels 0 and 3. This means that full shed and full exposure to sun have a negative impact on nest survival. Light (level 1) and moderate (level 2) canopy closure should be preferred for future transplantations in southern Turkey. Although landform, slope, and substrate do not seem to be among the most important factors in transplantation success, mid-hill, average slopes, and fallen trunks could be more advantageous in cases where other options are available in the same parameter classes.

## 5. Conclusions

In the present study, we showed that classification methods (i.e., decision tree and naïve Bayes) can be used in not only classifying but also predicting red wood nest transplantation success. This approach can assist practitioners in planning nest transplantation activities. In particular, the software we used for Bayesian analyses (Orange) is easy to use, and once the data is provided, it has an interface that can be rapidly manipulated to give success predictions for a desired set of habitat parameters. It should be noted that precision of this approach is highly dependent on the size of the dataset used. The greater the size of the data collected from the study region is, the greater the ability of the approach will be to predict the transplantation success. It should also be kept in mind that this approach would work better at a local scale if the data is at the considered local scale, because several biotic or abiotic habitat parameters other than those measured in this study could be in action and have significant impact on the survival of red wood ant nests. We assumed that those parameters were constant over years in our model, but this could be different in each local scale, for which the model should be designed from scratch for each case. Last, but not least, we did not take climatic change into account. Future projections under local climatic change scenarios should be incorporated into the model for better predictions. Nevertheless, our results revealed transplantation success-related parameters, and they can help to understand how climatic change may affect future red wood ant nest transplantation efforts in the study region.

**Author Contributions:** Conceptualization, K.İ., Ö.B., A.S., and U.M.A.; methodology, K.İ., A.S., U.M.A., and Ö.B.; software, K.İ. and Ö.B.; field work, A.S., U.M.A., A.Y., and H.İ.Y.; writing—original draft preparation, K.İ. and Ö.B.; visualization, Ö.B. and K.İ.; funding acquisition, A.S. and U.M.A. All authors have read and agreed to the published version of the manuscript.

**Funding:** This research was funded by the Turkish General Directorate of Forestry, project number 19.4702/2016-2018.

**Acknowledgments:** We thank M. Nihat Aktaç for the taxonomic identification of red wood ants, and Adem Sinan Hınıs of the Turkish General Directorate of Forestry for the annual red wood ant transplantation data in Turkey.

**Conflicts of Interest:** The authors declare no conflict of interest.

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
