# Peer review of "Nest Survival and Transplantation Success of Formica rufa (Hymenoptera: Formicidae) Ants in Southern Turkey: A Predictive Approach"

_forests, doi:10.3390/f11050533_

Round 1
Reviewer 1 Report
This manuscript “Nest survival and transplantation success of Formica rufa (Hymenoptera: Formicidae) ants in southern Turkey: A predictive approach” investigates how the success in nest transplantation of Fomica rufa is affected by habitat parameters in southern Turkey. The possibility of predicting transplantation success is investigated as well.
I think that the paper is interesting and that the investigations are well conducted, and overall I think it is suitable for the journal. However, I have some comments to be addressed before accepting this article.
Lines 44-45: It would be useful for the reader to have some specific information about the use of Formica rufa as biocontrol agent
Line 95: What is the difference between “active” and “successful” and “abandoned” and “unsuccessful” transplantations?
Line 112: please provide italic for “sensu stricto”
Lines 114-115: it is not clear why nest height and nest radius parameters were discarded from the analyses. Why it was not possible to use these parameters for the nests that were not destroyed? Were they too few for the analyses? Please explain better.
Line 117: is there something missing after “radius”?
Lines 174-175: please add statistical analyses results
Line 325, 332, 341, etc.: In the Discussion section the Authors do not compare their results to the data of the literature.
Author Response
“This manuscript “Nest survival and transplantation success of Formica rufa (Hymenoptera: Formicidae) ants in southern Turkey: A predictive approach” investigates how the success in nest transplantation of Formica rufa is affected by habitat parameters in southern Turkey. The possibility of predicting transplantation success is investigated as well.
I think that the paper is interesting and that the investigations are well conducted, and overall I think it is suitable for the journal. However, I have some comments to be addressed before accepting this article.”
Point 1. Lines 44-45: It would be useful for the reader to have some specific information about the use of Formica rufa as biocontrol agent
Response 1. We provided specific information regarding to the forest pests whose numbers have been reported to decrease as a result of Formica rufa attacks. Please see Lines 46-51.
Point 2. Line 95: What is the difference between “active” and “successful” and “abandoned” and “unsuccessful” transplantations?
Response 2. We completed the missing information. Please see Lines 99-100.
Point 3. Line 112: please provide italic for “sensu stricto”
Response 3. Done. Please see Line 121.
Point 4. Lines 114-115: it is not clear why nest height and nest radius parameters were discarded from the analyses. Why it was not possible to use these parameters for the nests that were not destroyed? Were they too few for the analyses? Please explain better.
Response 4. The number of destroyed nests were significantly high. And we could not use different sets of parameters for each nest because we looked for combined effects of these parameters. Therefore, all nests must have the same sets of parameters. For this reason, there were two options: (1) to discard these nests completely and to use all parameters for other nests including height and radius, or (2) To discard height and radius data and to include the destroyed nests. We preferred the second one because we thought that information related to the other parameters are more important than radius and height for nest survival and we did not want to lose these parameters coming from destroyed nests. But we will deposit the entire data (including height and radius) for future analyses. We added a sentence to explain this in Lines 124-125.
Point 5. Line 117: is there something missing after “radius”?
Response 5. A name of the repository where the data used in this study will be uploaded following the acceptance of the manuscript for publication. Please see Line 128.
Point 6. Lines 174-175: please add statistical analyses results
Response 6. Done. Please see Line 201-202.
Point 7. Line 325, 332, 341, etc.: In the Discussion section the Authors do not compare their results to the data of the literature.
Response 7. We added more references (those we used in Introduction) to the already present ones. Please see Lines 368-369, Lines 380-381, and Lines 390-391. But the scientific field of biocontrol by using Formica rufa is not as rich as other biocontrol topics; and among those even less reports the topics of F. rufa habitat parameter and transplantation. We tried to cover all available publications, and moreover we tried to rescue some of the publications in Turkish from grey literature, particularly in the Introduction.
Reviewer 2 Report
This is an interesting attempt to develop multidimensional models to predict the success of an insect as a function of a series of site characteristics. The data set is presented as a series of univariate variables (Fig. 3), and the results of the main variables are also discussed independently (L325-351). This is OK, but these variables presumably interact, which I imagine the statistical models also incorporate. The results of some of these interactions are presented, but only as vignettes (Figs. 4 and 5). The paper would benefit greatly by including some graphics that show probability of success as a contour plot or 3-D surface vs. pairs of important variables (e.g., altitude vs. aspect). Observed data could be superimposed on such a graphic to give the reader an impression of how the predictions match the data. Many models were run, but it is not clear to me if they produce similar predictions or whether some are better than others, and if so, why. Given the amount of work done, it seems that something should be learned from it and shared with the reader.
Introduction
What pests are meant to be controlled by this ant?
What is the native geographic range of this insect? Are they associated with a particular type of forest? Are all the releases outside of its known range?
Methods
L110 give some quantitative measure of the 4 canopy classifications -- enough information for someone to replicate the methods in another study.
L110 give some quantitative measure or more specific description of the 5 land form classifications so that this could be replicated.
L157 " we tested performance of each algorithm" -- I see that performance measures are presented in Table 1, but is this the same as "testing" the algorithms?
L158 The performance criteria Accuracy, Precision, Sensitivity, Specificity, F1 Score, and Area Under the 158 Curve (AUC) are listed, but without description of what they mean and their importance, nor is this described in the Results or Discussion sections.
Results
I am not experienced using these statistical tools. However, it is not clear what was learned from using the four different DT methods. Do they more or less agree in their classifications? Would the authors recommend which one to use in the future?
Table 3 appears to cherry pick the highest predictions from the various algorithms, but is there any consensus between the algorithms? If not, then how reliable are the predictions? How does Accuracy and Precision relate to these predictions?
The transplantation sites were further south from the origins, and the ants are probably adapted to sites that have similar environmental conditions such as temperature and moisture, so it makes sense that they would do better at higher elevations (or higher than those in Central Turkey) and with N to E aspect or with more canopy closure for shade. However, the results in Table 3 appear to indicate higher success at lower elevations.
I presume that the derived algorithms can be used to calculate the probability of success at any combination of parameters that were modelled; however, this is not available to the reader. Is there a way to graphically present these in multidimensional graphics (e.g. contour plots in 2 or 3 dimensions)? Otherwise, is the equation available so that one could calculate the probability for a given combination of parameter values?
Discussion
It is not clear how forest managers can directly benefit from this study because it does not present the probabilities calculated for the range of the various parameters that were studied. It would be useful to make this available as graphics, tables or equations so that someone can see the probability for whatever site characteristics they are interested in. Is the Nomogram a tool that could be available on the internet for users to operate?
L329 "we found relatively more active nests at higher sites than lower sites"; however Table 3 predicts high success at moderate elevations, considering that sites up to 1900 m were studied. Perhaps it is not clear what is meant by 'high' elevation. Consider that a graphic plotting success vs elevation and another parameter (e.g. aspect or canopy) would clearly show patterns.
Author Response
“This is an interesting attempt to develop multidimensional models to predict the success of an insect as a function of a series of site characteristics. The data set is presented as a series of univariate variables (Fig. 3), and the results of the main variables are also discussed independently (L325-351). This is OK, but these variables presumably interact, which I imagine the statistical models also incorporate. The results of some of these interactions are presented, but only as vignettes (Figs. 4 and 5). The paper would benefit greatly by including some graphics that show probability of success as a contour plot or 3-D surface vs. pairs of important variables (e.g., altitude vs. aspect). Observed data could be superimposed on such a graphic to give the reader an impression of how the predictions match the data. Many models were run, but it is not clear to me if they produce similar predictions or whether some are better than others, and if so, why.
Given the amount of work done, it seems that something should be learned from it and shared with the reader.”
Point 1. What pests are meant to be controlled by this ant?
Response 1. We provided specific information regarding to the forest pests whose numbers have been reported to decrease as a result of Formica rufa attacks. Please see Lines 46-51.
Point 2. What is the native geographic range of this insect?
Response 2. We provided native geographic range of Formica rufa. Please see Lines 40-41.
Point 3. Are they associated with a particular type of forest?
Response 3. We provided the type of forest (temperate coniferous forests) where Formica rufa lives. Please see Line 41.
Point 4. Are all the releases outside of its known range?
Response 4. Formica rufa releases have been undertaken both in and outside of its native range. We provided the information in Line 46.
Point 5. L110 give some quantitative measure of the 4 canopy classifications -- enough information for someone to replicate the methods in another study.
Response 5. We provided quantitative measures for the canopy classifications. Please see Lines 116-117.
Point 6. L110 give some quantitative measure or more specific description of the 5 land form classifications so that this could be replicated.
Response 6. We provided quantitative measures for the canopy classifications. Please see Lines 117-120.
Point 7. L157 "we tested performance of each algorithm" -- I see that performance measures are presented in Table 1, but is this the same as "testing" the algorithms?
Response 7. Actually, we tested the performances of the algorithms but not the algorithms themselves. Therefore, the test results presented in Table 1 are the results of performance tests for DT and NB done by using performance criteria.
Point 8. L158 The performance criteria Accuracy, Precision, Sensitivity, Specificity, F1 Score, and Area Under the 158 Curve (AUC) are listed, but without description of what they mean and their importance, nor is this described in the Results or Discussion sections.
Response 8. We explained better now the performance criteria, how and why there used in Lines 162-180.
Point 9. I am not experienced using these statistical tools. However, it is not clear what was learned from using the four different DT methods. Do they more or less agree in their classifications? Would the authors recommend which one to use in the future?
Response 9. As explained better now in the manuscript, the reason to use different algorithms in data mining studies is to select the algorithm that best suits the target variable. This depends largely on the data and should be tested for each data set. Therefore a recommendation for others to use in future studies would not be valid. Through performance testing we found all performances high and concluded that all algorithms could be used for our data set (but only for it) to predict transplantation success. We tried to bring some clarity by modifying the sentences that explain why four different DT methods were used and how they performed. Please see Lines 162-180, and Line 272.
Point 10. Table 3 appears to cherry pick the highest predictions from the various algorithms, but is there any consensus between the algorithms? If not, then how reliable are the predictions? How does Accuracy and Precision relate to these predictions?
Response 10. These are very important questions related to the nature of the statistical approach we used in our study and they helped us a lot to improve the way we explain our results. First of all, Accuracy and Precision (and other performance criteria) were used to test performances of the algorithms. They showed that each algorithm has high performances and could be used in prediction. We explained this more clearly now in the manuscript (see the previous answers). Second, there is no need to a consensus between the algorithms. Each of them use a different approach to partition the continuous data. As a result, for example, one algorithm can find higher transplantation success under 1420 m, while the other partitions it as 1371 m < Altitude < 1609 m but adds another parameter such as canopy closure level 1. The two results are not mutually exclusive and they are different bits of information related to the data. The same logic also applies for comparison between DT and NB. We tried to avoid to give an impression like we aimed to compare the two classification methods. Because it would not be a correct objective. The two methods are more complementary to each other because they work differently. Although the same data set is provided to both, DT selects the most explanatory part of the data and do the prediction according to this part, whereas NB uses the entire data to create predictions. Thus the two methods result in probabilities of different scenarios. And different scenarios cannot be compared. They can only be used as informative guides. These were explained in different parts of the manuscript to make it clearer. Please see Lines 162-164, Lines 185-187, Lines 194-197, Lines 290-293, and Lines 295-296.
Point 11. The transplantation sites were further south from the origins, and the ants are probably adapted to sites that have similar environmental conditions such as temperature and moisture, so it makes sense that they would do better at higher elevations (or higher than those in Central Turkey) and with N to E aspect or with more canopy closure for shade. However, the results in Table 3 appear to indicate higher success at lower elevations.
Response 11. The comment on the adaptation of transplanted ants to higher altitudes was one of the points we wanted but apparently failed to state clearly in the previous version of the manuscript. Now we improved it with the sentence in Lines 374-377. On the other hand, Table 3 does not indicate higher success at lower elevations. The probability of a nest being at a lower altitude but being more successful than a nest at a higher altitude is related to the other parameters. For example, please check the table below. It is just another example of the outcomes of our analyses and it shows that altitude positively effects the success. However these are simple probabilities. They just give an idea by ignoring other parameters. Therefore, altitude alone have a positive effect on transplantation success. But intervening effects of other parameters can change this. Our focus was to show this combined effect. And we tried a better explanation in Lines 290-293. We also tried to make Table 3 easier to follow by providing a longer explanation in the legend and more references to the in the text.
Algorithms |
Altitude |
Probability of Success |
C&RT |
>1420 |
%77.65 |
Chaid |
>1609 |
%82.74 |
Quest |
>1493 |
%80.89 |
C5.0 |
>1538 |
%82.71 |
Point 12. I presume that the derived algorithms can be used to calculate the probability of success at any combination of parameters that were modelled; however, this is not available to the reader. Is there a way to graphically present these in multidimensional graphics (e.g. contour plots in 2 or 3 dimensions)? Otherwise, is the equation available so that one could calculate the probability for a given combination of parameter values?
Response 12. As the number of possible probabilities is extremely high, a comprehensive display of all probabilities is not possible. Instead, the nomogram we provided at Figure 5 is more useful. Anyone who downloads the Orange Data Mining software can test their own data easily. The nomogram is an interactive interface ready to show all possible probabilities found from the data and the user can view all of them by simply sliding the cursor. We described this in the manuscript to make it clearer. Please see Lines 403-405.
Point 13. It is not clear how forest managers can directly benefit from this study because it does not present the probabilities calculated for the range of the various parameters that were studied. It would be useful to make this available as graphics, tables or equations so that someone can see the probability for whatever site characteristics they are interested in. Is the Nomogram a tool that could be available on the internet for users to operate?
Response 13. Please see the previous answer.
Point 14. L329 "we found relatively more active nests at higher sites than lower sites"; however Table 3 predicts high success at moderate elevations, considering that sites up to 1900 m were studied. Perhaps it is not clear what is meant by 'high' elevation. Consider that a graphic plotting success vs elevation and another parameter (e.g. aspect or canopy) would clearly show patterns.
Response 14. Please see the previous answers.
Round 2
Reviewer 1 Report
I thank the authors for their replies to the comments on the submitted version. They have done a very good revision from their earlier version, and in particular all the comments and suggestions have been addressed. I have no further comments to add.
This manuscript is a resubmission of an earlier submission. The following is a list of the peer review reports and author responses from that submission.